# Exploring the Diversity of the Thioredoxin Systems in Cyanobacteria

**DOI:** 10.3390/antiox11040654

**Published:** 2022-03-28

**Authors:** Manuel J. Mallén-Ponce, María José Huertas, Francisco J. Florencio

**Affiliations:** 1Instituto de Bioquímica Vegetal y Fotosíntesis, Universidad de Sevilla-CSIC, Américo Vespucio 49, 41092 Sevilla, Spain; floren@us.es; 2Departamento de Bioquímica Vegetal y Biología Molecular, Facultad de Biología, Universidad de Sevilla, Profesor García González s/n, 41012 Sevilla, Spain

**Keywords:** redox regulation, thioredoxin, thioredoxin reductase, cyanobacteria, oxidative stress, metabolism, evolution

## Abstract

Cyanobacteria evolved the ability to perform oxygenic photosynthesis using light energy to reduce CO_2_ from electrons extracted from water and form nutrients. These organisms also developed light-dependent redox regulation through the Trx system, formed by thioredoxins (Trxs) and thioredoxin reductases (TRs). Trxs are thiol-disulfide oxidoreductases that serve as reducing substrates for target enzymes involved in numerous processes such as photosynthetic CO_2_ fixation and stress responses. We focus on the evolutionary diversity of Trx systems in cyanobacteria and discuss their phylogenetic relationships. The study shows that most cyanobacteria contain at least one copy of each identified Trx, and TrxA is the only one present in all genomes analyzed. Ferredoxin thioredoxin reductase (FTR) is present in all groups except *Gloeobacter* and *Prochlorococcus*, where there is a ferredoxin flavin-thioredoxin reductase (FFTR). Our data suggest that both TRs may have coexisted in ancestral cyanobacteria together with other evolutionarily related proteins such as NTRC or DDOR, probably used against oxidative stress. Phylogenetic studies indicate that they have different evolutionary histories. As cyanobacteria diversified to occupy new habitats, some of these proteins were gradually lost in some groups. Finally, we also review the physiological relevance of redox regulation in cyanobacteria through the study of target enzymes.

## 1. Distribution of Cyanobacterial Thioredoxins

In all oxygenic photosynthetic organisms, several enzymes involved in CO_2_ fixation and many other processes are activated by the reduction of one or several disulfide bridge(s) formed between cysteines. In most cases, this reduction is carried out by thioredoxins (Trxs), which receive reducing equivalents from photosynthetic electron transport. Trxs are characterized by their low molecular mass, approximately 12–14 kDa, characteristic fold, and highly conserved active site (WC(G/P)PC). As discussed below, the first evidence for the role of Trxs in these organisms comes from studies of the activation of fructose-1,6-bisphosphatase (FBPase), which is involved in the Calvin–Benson–Bassham cycle (CBB). Subsequently, two Trxs named Trx-*m* and Trx-*f* were identified in plant chloroplasts [1,2], although multiple whole-genome sequencing of plants, algae, and cyanobacteria revealed a high diversity of Trxs [3,4,5,6]. Plastids contain the *f*-, *m*-, *x*-, *y*-, and *z*-type Trxs [7], three of which are conserved in cyanobacteria (*m*-, *x*-, and *y*-types) [4]. The activity of two Trxs was first described in the cyanobacteria *Anabaena* sp. PCC 7120 and *Synechococcus* sp. PCC 6301 [8,9,10,11]. Subsequently, the increase in the number of sequenced genomes allowed the identification of four types of Trx in cyanobacteria [4]. The three Trxs conserved among cyanobacteria and eukaryotic chloroplasts are *m*-type (also known as TrxA), *x*-type (TrxB), and *y*-type (TrxQ) Trxs. The fourth Trx, named TrxC, is unique to cyanobacteria. The initial study established that large cyanobacterial genomes contain more Trxs than smaller ones, although only 20 cyanobacteria were analyzed [4]. Recently, several cyanobacterial genomes have been sequenced and added to the IMG and NCBI databases. To explore whether Trxs are widespread among cyanobacteria, we obtained the amino acid sequences of Trxs from the IMG and NCBI databases and aligned them for phylogenetic analysis (ncbi.nlm.nih.gov and img.jgi.doe.gov, accessed date: 23 February 2022). We also included their closest living non-photosynthetic relatives, Melainabacteria (formerly Vampirovibrionia), Sericytochromatia, and Margulisbacteria. The resulting unrooted maximum likelihood was used to classify Trxs into five clades, corresponding to A-, A3-, B-, Q-, and C-type Trxs (Figure 1 and Appendix A). The TrxA (*m*-type) clade contains at least one TrxA from all cyanobacterial genomes analyzed (Figure 1). This clade includes the *Synechocystis* sp. PCC 6803 (hereafter *Synechocystis)* TrxA (*slr0623*) and *Anabaena* (also referred to as *Nostoc*) TrxA1 orthologue (*alr0052*). A small monophyletic group includes paralogues present in filamentous cyanobacteria such as *Anabaena* TrxA2 (*all1866*) (Figure 1 and Appendix A). *Anabaena* TrxA2 (*all1866*) is 75% identical to *Anabaena* TrxA1 (*alr0052*). This small group also contains paralogues in simple filamentous cyanobacteria such as *Pseudanabaena* spp. or *Leptolyngbya* spp. Sequence analysis of TrxA1 and TrxA2 from several cyanobacteria indicates that there are only five unique amino acids in TrxA2 relative to TrxA1 (Appendix A). Therefore, phylogenetic analysis and rooting of these paralogues show that they probably arose from an ancient duplication before the origin of multicellularity. Furthermore, phylogenetic analysis revealed that cyanobacterial TrxA has orthologues in its closest living non-photosynthetic relatives. Surprisingly, a clade is observed far from the TrxA clade and includes the third TrxA of *Anabaena* (*all2367*) (Figure 1, brown). This clade, which we refer to as TrxA3, also includes paralogues distributed in various genomes, including early-branching cyanobacterial lineages such as *Pseudanabaena biceps* PCC 7429, *Acaryochloris marina* MBIC1107, or *Anthocerotibacter panamensis,* a new species of Gloeobacteria recently described [12]. In *Anabaena*, TrxA3 is only 45% and 44% identical to TrxA1 and TrxA2, respectively.

TrxB is present in 310 of the 416 cyanobacterial genomes analyzed but is absent in known basal lineages (*Gloeobacter, Pseudanabaena, Thermosynechococcus,* and *Acaryochloris* spp.) (Figure 1 and Appendix A). Instead, the molecular analysis identified a TrxB orthologue in the early-branching cyanobacterium *Gloeomargarita lithophora*. On the other hand, TrxQ is present in 287 cyanobacterial genomes and is present in known basal lineages (*Pseudanabaena*, *Thermosynechococcus*, and *Acaryochloris* spp.) (Figure 1 and Appendix A). In addition, a new member of Gloeobacteria, *Anthocerotibacter panamensis*, contains a TrxQ orthologue. Finally, TrxC is only found exclusively in cyanobacteria and is present in 289 cyanobacterial genomes (Figure 1 and Appendix A). It is also absent in *Gloeobacter* and *Pseudanabaena* spp. In contrast to this, it is present in *Thermosynechococcus* spp. All three Trxs are missing in *Prochlorococcus* spp.

Co-occurrence patterns reveal that most cyanobacteria contain at least one copy of each Trx (137 out of 416 genomes) or of each Trx except TrxA3 (112 out of 416 genomes) (Figure 2A and Appendix A). Furthermore, TrxA is the only Trx present in 70 cyanobacterial genomes, including all *Prochlorococcus* and *Gloeobacter* spp. According to different studies on *Synechococcus* and *Synechocystis*, TrxA is the only essential Trx for survival under photoautotrophic and heterotrophic growth conditions [13,14,15]. The TrxA group showed short branches (Figure 1), reflecting that the amino acid substitution rate was low between orthologues. Sequence conservation among different Trxs shows that several patches of highly conserved residues exist in the TrxA group (Appendix A). In contrast, branches of the other groups are much longer, and residues and patches are less conserved.

Mapping the primary sequences of Trxs on the structural models reveals a conservation pattern around the active site (Figure 2B). However, the TrxA group is the only one with all residues conserved in this region. This high degree of conservation around the active site reflects the need to maintain natural integrity and function. Specifically, conserved regions contribute to the binding and reduction of their target proteins, as well as the reduction of TrxA by thioredoxin reductases. Both interactions occur with high specificity, which could explain the essential role of this Trx. Interestingly, the specific amino acids of TrxA2 versus TrxA1 are located on the opposite side of the active site in *Anabaena* (Figure 2B and Appendix A), which could not explain the specific functions of TrxA2. In this way, the redox regulation by TrxA was initially fulfilled with a single copy, and duplication of the TrxA gene could serve as a backup of essential functions. Furthermore, new target proteins were added during cyanobacteria diversification events, including certain specificity in some functions [16].

## 2. Diversity and Evolution of Cyanobacterial Thioredoxin Reductases

Ferredoxin:thioredoxin reductase (FTR), which links photosynthetic electron transport to Trx-based regulation, was discovered more than 40 years ago in chloroplasts and cyanobacteria [2,9]. FTR is a heterodimeric enzyme composed of an approximately 13 kDa catalytic subunit (FTRC), which contains a 4Fe-4S cluster and a redox-active site, and a variable subunit (FTRV) with a molecular mass ranging from 7 to 13 kDa, which may protect the 4Fe-4S cluster of FTRC from oxidative inactivation [16,17,18]. Crystallographic structures showed that ferredoxin (Fd) and Trxs interact exclusively with the catalytic subunit of FTR [19,20,21]. The structure of the FTR-Trx-*m* complex was solved, and the Trx-*m* residues that specifically interact with FTRC were identified [20,21]. These residues are widely conserved in cyanobacterial TrxA (Appendix A).

FTRC is found in a variety of bacteria and eukaryotes, while FTRV is exclusive to oxygenic photosynthetic organisms [16]. FTRC homologs were detected in bacteria with deep phylogenetic roots, suggesting an origin from microaerophilic bacteria that use Trx to regulate CO_2_ fixation by the reverse citric acid cycle [16,22]. Our bioinformatic analysis shows that the FTR complex is present in most cyanobacteria groups (Appendix A). FTR is only absent in *Prochlorococcus* and *Gloeobacter* spp. Furthermore, neither FTRC nor FTRV are identified in the closest living non-photosynthetic relatives of cyanobacteria.

Analysis of cyanobacterial genomes initially distinguished several types of NADPH-dependent thioredoxin reductase (NTR) as an alternative to the FTR complex. Recent structural studies have analyzed these possible NTRs and identified a new TR and a diflavin disulfide oxidoreductase. A few years ago, structural analysis revealed TR in some cyanobacteria such as *Gloeobacter* spp. [23], which was initially called deeply rooted thioredoxin reductase (DTR) and later renamed ferredoxin flavin-thioredoxin reductase (FFTR) [24]. The FFTR family was initially described in the nitrogen-fixing anaerobe *Clostridium pasteurianum* [25]. These proteins are evolutionarily related to prokaryotic NTR, but lack an NADPH-binding site [26]. Similar to FTR, FFTRs are reduced by Fd [25,26]. In some cyanobacteria such as *Gloeobacter* and *Prochlorococcus* spp., FFTR is the only enzyme with TR activity [23]. Another protein evolutionarily related to NTR was also recently described and named diflavin-linked disulfide oxidoreductase (DDOR) [27]. DDOR is a homodimer that contains a unique structural feature of two flavin cofactors bound to each subunit, one of which also contains a highly conserved CxxC motif. However, DDOR lacks an NADPH-binding site and has modifications at the Trx-binding site of the NTR family [27]. Although the mechanism by which DDOR functions is still incomplete, in vitro analysis suggested that glutathione may act as a physiological electron donor [27,28]. Further experiments are required to corroborate this possibility and identify possible acceptors for DDOR. Furthermore, typical NTR was identified in some cyanobacteria such as *Anabaena* [4], and further analysis revealed that it was specifically reduced to a Trx encoded downstream of the *ntr* gene [29], as will be discussed below. Finally, some cyanobacteria and chloroplasts contain a particular form of NTR called NTRC, with a Trx module fused to its C-terminus [4,30]. Based on the finding that NTRC is a very efficient reductant of 2-Cys peroxiredoxin (2-Cys Prx), an antioxidant role for this enzyme was initially proposed in plant chloroplasts [31]. In line with this, in vitro and in vivo analyses showed that NTRC is the main electron donor for 2-Cys Prx in *Anabaena* [32,33,34,35].

In recent years, the number of available cyanobacterial genome sequences has quadrupled, including genomic sequences of strains from diverse habitats. To explore the distribution pattern and diversity of other TR-related enzymes, putative sequences of cyanobacterial FFTRs, DDORs, NTRs, and NTRCs were obtained from the IMG and NCBI databases and aligned for phylogenetic analysis. TR-related enzymes identified in Melainabacteria, Sericytochromatia, Margulisbacteria, and *E. coli* were also included in the analysis. The resulting unrooted maximum likelihood was used to classify these proteins into three main clades, corresponding to NTR/NTRC, FFTR, and DDOR (Figure 3A and Appendix A). In addition, molecular phylogeny identified other NTRs in some cyanobacteria such as *Anabaena* sp. PCC 7524, which formed nodes with the NTRs of Melainabacteria spp. or *E. coli* (orange). These NTRs, which we refer to as NTR2, retain the Trx-binding site of the NTR family (Figure 3B). Although the function of these proteins is unknown, they may arise from horizontal gene transfer between cyanobacteria and other bacterial phyla.

The NTR/NTRC clade includes two clearly distinguishable subclades (Figure 3A). On the one hand, a large subclade contains the well-characterized Anabaena NTRC (*all0737*) (NTRC1 in Figure 3A). Interestingly, this subclade also includes NTR sequences from *Pseudanabaena* spp. and *Synechococcus* sp. PCC 7502, which we refer to as NTR1 (blue within the NTRC1 subclade in Figure 3A). *Pseudanabaena* spp. contain NTRC or NTR1 and form a monophyletic group (Appendix A), suggesting a common origin. On the other hand, a large subclade includes the NTRC orthologues of *Prochlorococcus* spp. and evolutionarily related *Synechococcus* spp. (NTRC2 in Figure 3A). Furthermore, other NTR1 sequences are identified in other cyanobacteria, including *Gloeobacter kilaueensis* or *Anabaena* sp. PCC 7120, closely related to NTR sequences of the closest living non-photosynthetic relatives (blue).

NTR1 protein sequences lack a Trx binding site (Figure 3B and Appendix A), and some cyanobacteria such as *Anabaena* have a non-canonical Trx downstream of NTR1 (Figure 4). AnNTR1 is only reduced by this Trx [29], and we propose to name it TrxE (*alr2205*) (Pérez-Pérez and Florencio unpublished). Due to the presence of an atypical active site (YCPSC) in this Trx, this is not reduced by FTR [29]. Phylogenetic analysis indicates that TrxE is evolutionarily distant from the other Trxs (Figure 1). Furthermore, the alignment of NTR1/TrxE (a hypothetical chimeric protein) and *Anabaena* NTRC allows observing a high homology with the NAD(P)H- and Trx-domains, respectively (Appendix A), indicating that NTRC and NTR1/TrxE have a common origin from gene duplication. Interestingly, a gene neighborhood survey reveals an unexpected variety of genes that encode other redox proteins, including disulfide isomerases (PDI) (WCXXC motif), glutathione S-transferase (GST), CPYC-type glutaredoxin (GRX), and/or GRX-like (CPLC motif) (Figure 4A). Based on this finding, NTR1/TrxE and NTRC possibly originated as part of the antioxidant system. Just as NTRC specifically reduces 2-Cys Prx, NTR/TrxE could probably reduce other components of the antioxidant system.

The DDOR clade has a long internal stem and is separate from the other clades (Figure 3A). DDOR is present in 87 cyanobacterial genomes and is distributed between all groups except *Prochlorococcus*. All DDOR protein sequences share a redox-active CxxC site and a FAD-binding site (GxGxxG) with NTR protein sequences but lack NADPH- (GxGxxA/G) and Trx- (GR/KG and FF) binding motifs (Figure 3B and Appendix A). In *Synechocystis*, a DDOR mutant is highly sensitive to oxidative stress [36]. To test for putative target proteins and their functions, we also analyzed the most prevalent neighboring genes and observed a close relationship with a type II peroxiredoxin (PrxII) and a repressor (Fur)-type transcriptional regulator (PerR) (Figure 4B). PerR acts as an inducer of PrxII in response to oxidative stress [37]. Since the DDOR protein can receive electrons from glutathione [28], it could be part of a redox couple under stress conditions. The mechanism and target proteins of DDOR represent important issues for future studies.

A third clade, close to the DDOR group, is clearly distinguishable (Figure 3A). This clade contains all cyanobacterial FFTRs, with 95 genomes identified with an FFTR. All FFTR protein sequences share a redox active-site CxxC, a binding site for FAD (GxGxxG), and a Trx- (GR/KG and FF) binding site with the NTR type but lack the binding motif NADPH (GxGxxA/G) (Figure 3B and Appendix A). Although FFTR was not identified in the other groups analyzed, this enzyme was found in an anaerobic bacterium [23]. In cyanobacteria, all *Gloeobacter* and *Prochlorococcus* spp., which lack the FTR complex, contain an FFTR in their genome (Appendix A). Because both the *Gloeobacter* and *Prochlorococcus* groups have a single TrxA, the Trx-redox system in these organisms consists of Fd, FFTR, and TrxA. Despite this, both groups are taxonomically distant from each other, suggesting very different evolutionary histories, as will be discussed below.

## 3. Timing the Expansion of TRX and TR in Cyanobacteria

Different lines of geochemical evidence have identified the main oxygenation events observed during the early Earth. The largest accumulation of oxygen in the atmosphere is generally referred to as the Great Oxidation Event (GOE) [38], which occurred in the late Archean period [39,40]. During this period, cyanobacteria diversified in different habitats [41], contributing to increased diversification rates. The phylum is composed of two extant groups or classes, Phycobacteria and Gloeobacteria that diverged around the GOE (2 billion years ago). Phycobacteria comprise almost all species of cyanobacteria. Gloeobacteria spp. lack thylakoids, and photosynthesis occurs in the cytoplasmic membrane. Although initially composed of two species of *Gloeobacter*, recent studies have identified new species of Gloeobacteria [12,42]. The *Anthocerotibacter panamensis* member diverged from *Gloeobacter* spp. more than 1.4 billion years ago and differs from *Gloeobacter* spp. in key ways, such as the carotenoid synthesis pathway [12]. *Gloeobacter* spp. synthesizes carotenoids as other bacteria, whereas *Anthocerotibacter panamensis* uses the typical pathway of cyanobacteria and plastids. The thioredoxin reduction system in *Anthocerotibacter panamensis* is not the FFTR type as found in *Gloeobacter* spp. but is rather through the FTR complex, which is typical in most cyanobacteria (Figure 5). This implies that the FTR complex evolved in the most recent common ancestor of Cyanobacteria and not Phycobacteria. FTRC was incorporated from evolutionarily deeply rooted species by horizontal gene transfer, and FTRV was introduced to protect the Fe-S cluster [16]. The distinctiveness of *Gloeobacter* spp. is another product of lineage-specific reductive evolution. In contrast, FFTR is absent in the basal lineages except for *Gloeobacter* and *Acaryochloris* spp. (Figure 5). Because the common ancestor gave rise to basal lineages before diverging into two major groups (Macrocyanobacteria and Microcyanobacteria) [43] (Figure 5), FFTR may also have been present in the common shared ancestor of all extant cyanobacteria. Furthermore, all cyanobacterial FFTRs form a monophyletic group (Appendix A), indicating a single origin and excluding horizontal gene transfer events within Cyanobacteria.

Most FFTRs are predominantly found among members and close relatives of *Prochlorococcus* (Figure 5 and Appendix A). Genome studies have pointed to a reduction in genome size within *Prochlorococcus* spp. and the most closely related genus, *Synechococcus* [44,45,46,47]. In *Prochlorococcus*, natural selection has favored the loss of genes, except for the most necessary ones, because the marine environment is poor in elements such as N, P, or Fe [48,49,50,51]. Phylogenomic analysis using marker genes (e.g., 16S rRNA gene) has previously shown that *Prochlorococcus* and *Synechococcus* spp. evolved from filamentous ancestral cyanobacteria [43,52,53], suggesting that the FTR complex was likely lost during the evolution of a marine phytoplankton lifestyle. The loss of the FTR complex could be related to an iron deficiency in its natural habitat, which leads to an essential role for FFTR in these organisms. Interestingly, half of the FFTRs identified outside the *Prochlorococcus* group correspond to closely related *Synechococcus* spp. such as KORDI-100 or CC9902 (Appendix A). Unlike *Prochlorococcus* spp., all studied *Synechococcus* spp. contain the FTR complex and only some contain FFTR (Appendix A).

In the case of Trxs, *Anthocerotibacter panamensis* contains TrxA3 and TrxQ, in addition to TrxA, suggesting that the cyanobacteria ancestor probably had these Trxs (Figure 5). Furthermore, TrxB and TrxC are present in other basal lineages and were probably also present in the ancestor based on their position in the phylogeny (Figure 1 and Appendix A). Our molecular phylogeny, including the closest relatives of cyanobacteria, points to a deeper origin of all cyanobacterial Trxs. Cyanobacterial Trxs do not form a large monophyletic group with their close relatives, which excludes duplication events after the divergence of cyanobacteria. In contrast, TrxA orthologues of Cyanobacteria, Melainabacteria, Margulisbacteria, and Sericytochromatia form a monophyletic group (Figure 1), suggesting that the other Trxs were probably lost in non-photosynthetic species. In an alternative scenario, cyanobacterial ancestors acquired TrxA3, TrxQ, TrxB, or TrxC through horizontal gene transfer. A detailed analysis of the presence of these Trxs in other bacterial phyla reveals only weak sequence similarity in Bacteroidetes, Firmicutes, or Verrucomicrobia.

Finally, DDOR and NTRC do not interact with Trxs, but their origins may be closely related. Furthermore, both proteins are found in the most basal lineages such as *Gloeobacter* and *Anthocerotibacter panamensis,* respectively (Figure 5), suggesting that they were present in the ancestor of cyanobacteria. Regarding DDOR, we also found DDOR-like proteins in Melainabacteria and Margulisbacteria, indicating that the ancestral DDOR protein existed before the divergence of Margulisbacteria, the group most phylogenetically distant from this analysis. In the case of NTRC, its absence in *Gloeobacter* genomes suggested that it originated in Phycobacteria. However, analysis of *Anthocerotibacter panamensis*, which is the phylogenetical sister to *Gloeobacter* spp., reveals that it contains NTRC, NTR1, and TrxE sequences (Appendix A). This suggests that both NTRC and NTR1/TrxE are probably ancestral to all cyanobacteria (subsequently, NTRC and TrxE were lost in *Gloeobacter* spp.) (Appendix A). Consequently, the phylogenetic position of both proteins places the origin of NTRC, through the fusion of an NTR and a Trx, after the divergence of cyanobacteria (Appendix A).

In this way, DDOR and NTRC could have emerged to complement the TR/Trx system in a new scenario, because they appear to have acquired roles in response to oxidative stress [32,33,36]. The evolution of oxygenic photosynthesis led to an increase in O_2_. Since O_2_ is highly reactive, cyanobacterial ancestors had high selective pressure and developed very efficient antioxidant mechanisms. As cyanobacteria diversified to occupy new habitats [41,54], some groups lost DDOR and/or NTRC that initially allowed crown cyanobacteria to protect against oxidative stress.

## 4. Early Signs and In Vitro Approaches to Identify Functions of Thioredoxins

Evidence that cyanobacteria, like plants, contain Trxs was reported in the early 1980s [8,9,11]. One of these Trxs was identified as *m*-type Trx [11] and cloned from *Anabaena* sp. PCC 7119 [55]. In *Synechococcus* sp. PCC 6301 (*Anacystis nidulans*), *m*-type Trx was found to be essential for survival under photoautotrophic growth conditions [14]. This Trx was also shown to be essential under heterotrophic and photoautotrophic growth conditions in *Synechocystis* [13]. Subsequently, another Trx was identified and classified as an *x*-type Trx from *Anabaena* sp. PCC 7120 (hereafter *Anabaena*) [56]. The cyanobacterial *m*- and *x*-type Trxs are commonly referred to as TrxA and TrxB, respectively [57]. Additionally, early biochemical studies using purified Trxs provided in vitro evidence of redox-regulated enzyme activities (Table 1). These studies revealed that Trxs activate three CBB enzymes, FBPase [10,58,59,60], sedoheptulose-1,7-bisphosphatase (SBPase) [59,61], and phosphoribulokinase (PRK) [59]. Furthermore, it was shown that Trx deactivates glucose-6-phosphate dehydrogenase (G6PDH) [62,63], the first enzyme of the oxidative pentose phosphate (OPP) pathway. An additional regulatory component, namely OpcA, acts as an allosteric effector of G6PDH. In *Nostoc punctiforme ATCC* 29133, biochemical experiments showed that TrxA suppresses G6PDH activation by reducing OpcA [64]. More recently, in vitro analysis using purified recombinant Trxs, G6PDH, and OpcA showed that TrxA1 and TrxA2 specifically regulate the activity of G6PDH in *Anabaena* [65].

In the 1990s, evidence that Trxs play a role in a wide variety of cellular processes led to the search for new protein targets. For this purpose, several ingenious high-throughput screening procedures [65,66,67] were developed. Our group carried out a strategy for the identification of Trx target proteins using *Synechocystis* Trx mutants, in which an internal cysteine was replaced by serine [57,66,67]. Later, a similar proteomic study identified candidate target proteins in *Anabaena* heterocysts and vegetative cells [68]. Subsequently, the roles of Trxs, mainly TrxA, were extended to the regulation of numerous other proteins and functions in more recent biochemical studies (Table 1). TrxA serves as a reducing substrate for the elongation factors G [69] and Tu [70], suggesting that TrxA regulates the translational machinery in vivo. Other cellular processes that also appear to be regulated by TrxA are nitrogen fixation [29], glycogen synthesis [71], and the oxidative stress response [72]. In *Synechocystis* and *Anabaena*, proteomic studies identified several peroxiredoxins (Prxs) among target proteins [57,66,67,68]. Prxs are key players in antioxidant systems and are conserved between cyanobacteria and chloroplasts [73]. Subsequently, in vitro analysis of protein–protein interactions and enzymatic activities showed that all *Synechocystis* Prxs are Trx-dependent peroxidases [72]. TrxQ is the main electron donor for PrxII, PrxQ2, and 2-Cys Prx, although TrxA and TrxB can also act as electron donors. TrxA is the main donor for 1-Cys Prx while TrxA and TrxB can donate reducing equivalents to PrxQ1 [72].

**Table 1 antioxidants-11-00654-t001:** Trx target proteins described in vitro. Proteins are indicated with abbreviations for the name of the organism: *Synechocystis* TrxA (SynTrxA), *Synechocystis* TrxB (SynTrxB), *Synechocystis* TrxQ (SynTrxQ), *Anabaena* TrxA1 (AnTrxA1), *Anabaena* TrxA2 (AnTrxA2), *Nostoc muscorum* TrxA (NmTrxA), and *Microcystis Aeruginosa* TrxA (MaTrxA).

Cellular Processes	Target Protein	Regulation In Vitro by	References
CBB cycle	FBP/SBPase	SynTrxA	[10,58,59,60,61]
	PGK	SynTrxA	[74]
	CP12	AnTrxA1	[75,76,77,78,79]
	PRK	NmTrxA/MaTrxA	[59,79]
OPP pathway	G6PDH	SynTrxA/SynTrxB	[62,63]
	OpcA	AnTrxA1/AnTrxA2	[64,65]
Nitrogen fixation	NifU	AnTrxA1	[68]
Glycogen metabolism	AGP	SynTrxA	[71]
	PGM	SynTrxA	[67]
Antioxidant defense	2-Cys Prx	SynTrxA/SynTrxQ/AnTrxA1	[35,72]
	1-Cys Prx	SynTrxA/SynTrxQ	[72,80]
	PrxQ1	SynTrxA/SynTrxB	[72]
	PrxQ2	SynTrxA/SynTrxQ	[72]
	PrxII	SynTrxA/SynTrxQ/SynTrxB	[72,80]
Transcriptional regulation	RpaB	SynTrxA	[81]
	RpaA	SynTrxA	[81]
	ManR	SynTrxA	[82]
	RexT	AnTrxA2	[83]
	PedR	SynTrxA/SynTrxB	[84]
	FurA	AnTrxA	[85]
	GntR-like (Sll1961)	SynTrxA	[86]
Protein synthesis	EF-Tu	SynTrxA	[70]
	EF-G	SynTrxA	[69]

Other in vitro studies also suggested that Trxs are involved in the regulation of gene expression (Table 1). A redox-active transcriptional repressor of the *trxA2* gene, named RexT, was identified in *Anabaena* heterocysts. DNA binding activity of RexT is lost by the formation of an intramolecular disulfide bond under oxidative conditions, whereas the DNA binding activity is restored via the interaction with TrxA2 [83]. In *Synechocystis*, a small LuxR-type transcription factor, named PedR, was identified as an interacting partner of TrxA and TrxB [84,87]. A screening system that uses *E. coli* co-expression strains analyzed the interactions between Trxs and transcriptional factors of the OmpR family present in the *Synechocystis* genome [81]. This study identified three of them as new candidates for interaction with TrxA (RpaA, RpaB, and ManR). ManR functions as a repressor of the *mntCAB* operon that encodes a manganese transporter under non-stress conditions [82]. The response regulators RpaA and RpaB are master transcription factors in cyanobacteria that control physiology in light–dark cycles [88,89]. Fur (ferric uptake regulator), which is the master transcriptional regulator of iron homeostasis, could also be reduced by TrxA in *Anabaena* sp. PCC 7120 [85] Finally, a GntR-family transcriptional factor (Sll1961) involved in acclimation responses also interacts in vitro with TrxA [86]. The identification of Trx-interacting transcriptional factors suggests that Trx may be a key regulator of transcriptional regulation, which should be studied in the future.

## 5. The Role of Thioredoxins in Day-Night Cycles in Cyanobacteria

In recent years, genetic approaches have been used to investigate the biological significance of redox regulation and the in vivo roles of the different Trx isoforms. In *Synechocystis*, the analysis of mutants showed specific functions for TrxC, TrxB (*x*-type Trx in chloroplasts), and TrxQ (*y*-type Trx in chloroplasts) under different growth conditions [90,91]. All three mutants were shown to be viable under normal growth conditions, indicating a non-essential role of these Trxs in *Synechocystis*. However, the TrxB-deficient mutant was specifically affected during shifts from low to high light intensity, suggesting a role in adaptation to high light [90]. In contrast, the analysis of an *x*-type Trx knockout mutant in *Arabidopsis* exhibited altered redox homeostasis, although showing no growth phenotype [92]. The TrxQ-deficient mutant showed a growth phenotype under oxidative stress conditions in *Synechocystis* [90]. Interestingly, *Arabidopsis* mutants deficient in *y*-type Trx showed sensitivity to high light and drought stress [93,94], revealing their role as a major antioxidant in chloroplasts and tolerance to plant stress. Overall, TrxB and TrxQ appear to act as reducing substrates for the oxidative stress response from cyanobacteria to chloroplasts. More recently, the characterization of TrxC knockout mutants in *Synechocystis* and *Anabaena* revealed possible pathways regulated by this Trx [91,95]. Mutants in both cyanobacteria showed altered pigment content. In *Synechocystis*, the TrxC mutant showed an altered growth phenotype under low-carbon conditions compared to the wildtype. In *Anabaena,* the TrxC mutant exhibited changes in the relative abundance of proteins involved in amino acids and carbohydrate metabolism. Furthermore, quantitative proteomics also showed changes in detoxification-related proteins. Further experiments using these mutants are likely to unravel which proteins are targets of TrxB, TrxQ, and TrxC, providing more information on their specific functions.

In vivo functions of TrxA have been suggested to be more diverse than those of other Trxs. In *Synechocystis* and *Anabaena*, TrxA represents approximately 80-90% of the total Trx pool [4,29]. In the case of *Anabaena*, which has three TrxA isoforms, the amount of TrxA1 (Trx-*m*1) was significantly higher than that of the other isoforms. TrxA2 (Trx-*m*2) was undetectable, while TrxA3 (Trx-*m*3) was 20 times less abundant than TrxA1. Due to the high homology between TrxA1 and TrxA2 (Appendix A), the generation and characterization of TrxA1 and TrxA2 knockout mutants suggest that both isoforms could compensate each other for essential functions. The ∆*trx-m2* mutant strain growth was similar to that of the wildtype regardless of the nitrogen source, while that of the ∆*trx-m1* mutant strain was significantly suppressed in the absence of nitrate.

The OPP pathway is one of the pathways for carbon catabolism in cyanobacteria [96]. As some steps of this pathway can operate both to oxidize carbohydrates and to fix CO_2_, it is tightly regulated [97]. The first step of the OPP pathway is catalyzed by G6PDH, which is essential for nitrogen fixation and dark heterotrophic growth in *Nostoc punctiforme* [98]. In *Anabaena*, analysis of the in vivo redox state of OpcA, a G6PDH-activating protein [64], revealed that it is reduced under photosynthetic conditions in the presence of nitrate [65]. In contrast, the redox state of OpcA remained partially oxidized in the ∆*trx-m1* mutant strain, suggesting that TrxA1 is the main electron donor for OpcA (Table 2). Furthermore, OpcA is mainly oxidized under nitrogen-fixing conditions [29,65]. Filamentous cyanobacteria, such as *Anabaena,* fix nitrogen during the day [99,100], where nitrogenase complexes can be inactivated by molecular oxygen from photosynthesis. Differentiation of vegetative cells into heterocysts, where nitrogenase is found, allows nitrogen fixation to remain active [101]. This is because heterocysts are surrounded by a thick cell wall that prevents oxygen from entering [102]. Furthermore, heterocysts lack linear photosynthetic electron transport and carbon fixation, and are limited to heterotrophic metabolism [103]. Carbohydrates from vegetative cells are catabolized via the OPP pathway, where G6PDH catalyzes the first step and provides NADPH to reduce nitrogenase. Although TrxA1 and TrxA2 are reduced in light, target proteins are partially oxidized in heterocysts even under light conditions [29]. This is possibly due to the lack of linear photosynthetic electron transport and the limited levels of the FTR/Trx system.

To investigate the role of TrxA in *Synechocystis,* our group recently developed a strategy based on the low-level expression of TrxA [15]. We selected the arsenic-inducible *arsB* promoter because of undetectable levels of the *trxA* gene in the absence of the inducer. Furthermore, the absence of a specific ribosome binding site (RBS) resulted in significantly reduced levels of TrxA in the presence of the inducer. The new mutant strain, named STXA2, showed TrxA levels of 10% and only small phenotypic differences compared to the wildtype in the presence of the inducer. This indicates that even these reduced TrxA levels in the STXA2 strain are sufficient to support all essential TrxA functions. In contrast, the removal of the inducer resulted in large phenotypic changes. Photosynthetic analysis showed a strong limitation in the CBB cycle. Analysis of the in vivo redox state of dual-function fructose-1,6/sedoheptulose-1,7-bisphosphatase (FBP/SBPase) revealed that it remains mainly reduced under photosynthetic conditions in the wildtype strain, while it is mainly oxidized in the STXA2 mutant strain after inducer removal (Table 2). Both phosphatase activities operate in the regeneration stage of the CBB cycle, where they play a key role in cell growth and carbon metabolism [104,105]. Interestingly, the crystal structure of FBP/SBPase was resolved in *Synechocystis* and *Thermosynechococcus elongatus* [106,107]. Functional FBP/SBPase seems to be a tetramer, including an AMP-binding site and a disulfide bridge. Although site-directed mutagenesis and enzyme activities suggested an in vivo disulfide bridge between C75 and C99 [100], structure analysis appears to indicate an in vivo disulfide bridge between C75 and C85 [106,107].

The CBB cycle has a unique organization in photosynthetic organisms, where some enzymes have different evolutionary origins. Certain chloroplast enzymes are derived from the last common ancestor of eukaryotes [108]. Regarding the activities of SBPase and FBPase, chloroplasts have two separate enzymes that catalyze individual reactions [109,110], while all cyanobacteria contain the bifunctional enzyme FBP/SBPase [105]. Our data are an example of how Trx-mediated redox regulation of some cellular processes is as conserved in cyanobacteria as in chloroplasts, although the target proteins are evolutionarily different. However, other enzymes present in chloroplasts have a cyanobacterial origin, such as PRK and glyceraldehyde 3-phosphate dehydrogenase (GAPDH), another enzyme of the CBB cycle. PRK is the only enzyme that conserves redox-regulated cysteines that are essential for its activity. PRKs are dimeric, and each monomer contains two conserved pairs of cysteines in oxygenic photosynthetic organisms. Early studies showed that PRKs from plants and algae are redox-regulated by a pair of cysteines [111,112]. Cyanobacterial PRKs also contain the same cysteine pair [59,75] and recent structural studies have shown that they are the basis for Trx-mediated redox regulation [76,77,78]. The structures of the oxidized PRK from *Synechococcus* sp. strain PCC 6301 and the GAPDH/CP12/PRK complex of *Thermosynechococcus elongatus* BP-1 were solved and showed the formation of a disulfide bridge between cysteines 19 and 41 [76,77]. PRKs consume ATP to produce ribulose bisphosphate (RuBP), and the ATP-binding site is disrupted in oxidized PRK [77,78]. Furthermore, the small protein CP12 binds to PRK and GAPDH, leading to the inactivation of both enzymes [113,114]. Reduced CP12 is an intrinsically disordered protein in the light [115,116,117], while oxidized CP12 is stabilized by two disulfide bridges in the dark [118,119]. Oxidized CP12 interacts with GAPDH before binding to oxidized PRK [76,120]. In cyanobacteria, the dissociation of the GAPDH/CP12/PRK complex depends on the reduction of CP12 and PRK [59,76,79,121], because GAPDH lacks redox regulation. Future studies will be needed to identify specific Trx(s) for the redox regulation of CP12 and PRK in cyanobacteria. A recent study has shown how CP12 is mainly reduced in *Anabaena* vegetative cells, regardless of the nitrogen source [29]. In contrast, the in vivo redox state of CP12 was found to be oxidized in heterocysts in the absence of a nitrate source, even under light conditions. As discussed above, Trx-mediated redox regulation does not function in these specialized cells.

Most cyanobacteria have an anabolic metabolism during the day and a catabolic metabolism at night, where some regulatory processes are critical for their coordination. TrxA-mediated redox regulation appears to regulate the critical step between OPP and CBB cycle activities during these transitions (Figure 6). During the night, oxidized CP12 binds and inactivates GAPDH and PRK. Furthermore, FBP/SBPase and PRK are also oxidized in the dark. The initiation of glycogen degradation is essential during the dark period in cyanobacteria. Most of the released glucose is shunted directly to the OPP pathway to generate reducing power in the form of NAD(P)H. G6PDH is the first enzyme in the OPP pathway and is active in the dark, where its activator protein OpcA is oxidized (Figure 6). At the onset of light, the photosynthetic electron chain transfers reducing equivalents to the Trx system and results in Trx-mediated reduction of FBP/SBPase, PRK, and CP12, which also releases GAPDH and PRK, ultimately activating the CBB cycle [88]. Furthermore, OpcA is reduced under these conditions, and G6PDH is inactivated, partially inactivating OPP activity (Figure 6). Analysis of the *Synechocystis* mutant with low levels of TrxA revealed a specific role for TrxA in the light-sensitive reduction of FBP/SBPase [15]; however, CP12, PRK, and OpcA have only been shown to be specifically reduced in vitro by TrxA1 or TrxA2 in *Anabaena* [65,79,121]. Future studies are required to identify other Trx target proteins involved in cyanobacterial metabolism regulation in vivo and to analyze the possible role of other Trxs in addition to TrxA.

ROS are mainly produced by the photosynthetic electron transport chain and have beneficial effects on some processes such as iron acquisition or cell signaling [122,123]. Analysis of cyanobacterial PRX mutants revealed a severe growth retardation phenotype under different light intensities or H_2_O_2_ [34,37,80,124,125,126], providing an essential role for these proteins in the response to oxidative stress. In *Synechocystis*, a recent in vivo study showed that TrxA is the primary electron donor for 2-Cys Prx, which is essential to keep it in a reduced state under light conditions [15]. However, many cyanobacteria such as *Anabaena* contain NTRC, which transfers reducing equivalents more efficiently to 2-Cys Prx than TrxA [33]. *Anabaena* NTRC is similar to plant NTRC in its ability to reduce 2-Cys Prx [35]. Since neither *Anabaena* NTRC nor plant NTRC reduces *Synechocystis* 2-Cys Prx [35], two evolutionarily divergent strategies coexist to cope with oxidative stress [92]. Future in vivo studies will be necessary to resolve the role of NTRC and TrxA with respect to other Prxs and the function of other Trxs in antioxidant defense.

## 6. Conclusions

The study of the Trx system reveals great diversity to cope with redox control within the cyanobacterial world. Although previous research on Trx-mediated redox regulation mainly focused on the enzyme FTR, which links light to the regulation of target enzymes, this study provides insight into the origin and evolution of the Trx and oxidoreductases network recently described in cyanobacteria. We have found that two TRs, FTR and FFTR, coexisted in ancestral cyanobacteria although only FTR was widely spread throughout evolution, including in chloroplasts. Instead, some groups of cyanobacteria evolved from FFTR and TrxA as the only TRX system. Additionally, we have identified all Trxs and oxidoreductases such as DDOR and NTRC in basal lineages and it seems plausible that these were also linked to cyanobacteria early in evolution. Niche adaptation promoted evolutionary diversification of the different cyanobacterial lineages and resulted in the loss of some of these components. Further studies are required to formally establish the importance of these redox proteins and clarify their role in the redox regulatory network. Moreover, it will allow us to understand how cyanobacteria have adapted to different environmental conditions.

## Figures and Tables

**Figure 1 antioxidants-11-00654-f001:**
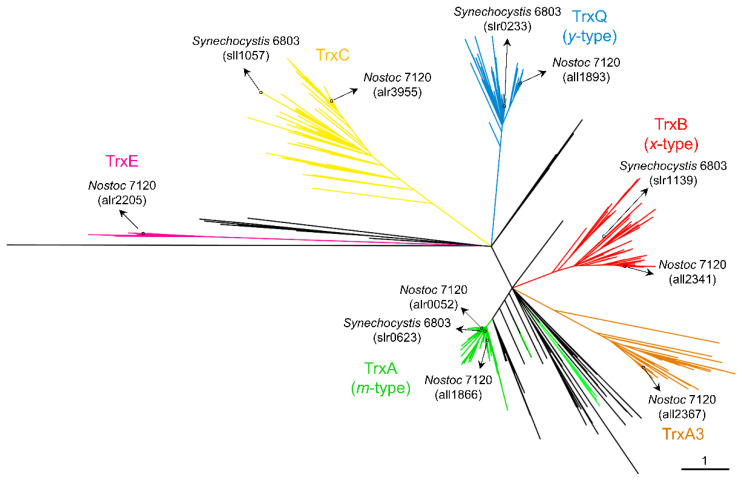
Phylogenetic distribution of cyanobacterial thioredoxins. Unrooted tree of full-length Trx amino acid sequences identified in the IMG and NCBI databases. Branches are colored to represent proteins found in the different subclades: TrxA1/2 (green), TrxA3 (brown), TrxB (red), TrxQ (blue), TrxC (yellow), and TrxE (purple). The scale bar represents the number of substitutions per site. Branches related to Melainabacteria, Sericytochromatia, and Margulisbacteria are colored black.

**Figure 2 antioxidants-11-00654-f002:**
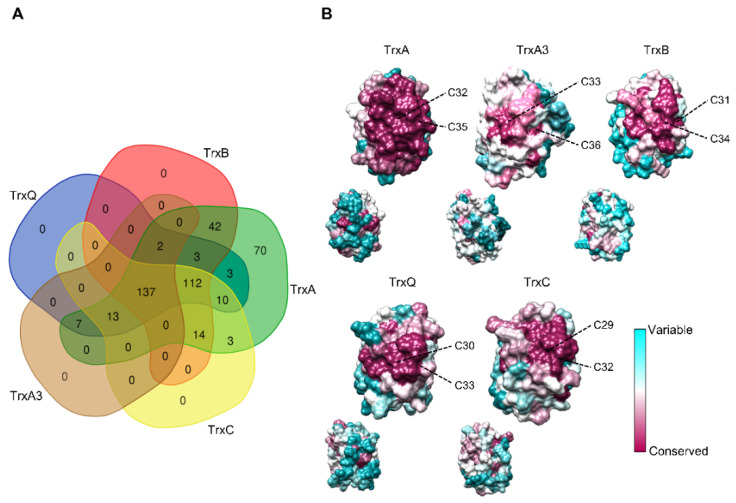
Distribution and conservation of cyanobacterial Trxs. (**A**) Venn diagram showing co-occurrence of Trxs in cyanobacteria analyzed. The specific cyanobacteria for each category are available in the Appendix A. (**B**) Residue-based conservation score plotted for models of TrxA1/2, TrxA3, TrxB, TrxQ, and TrxC. Residues are colored according to conservation scores ranging from 1 (cyan, least conserved) to 9 (purple, most conserved). The active-site cysteines are indicated for each model.

**Figure 3 antioxidants-11-00654-f003:**
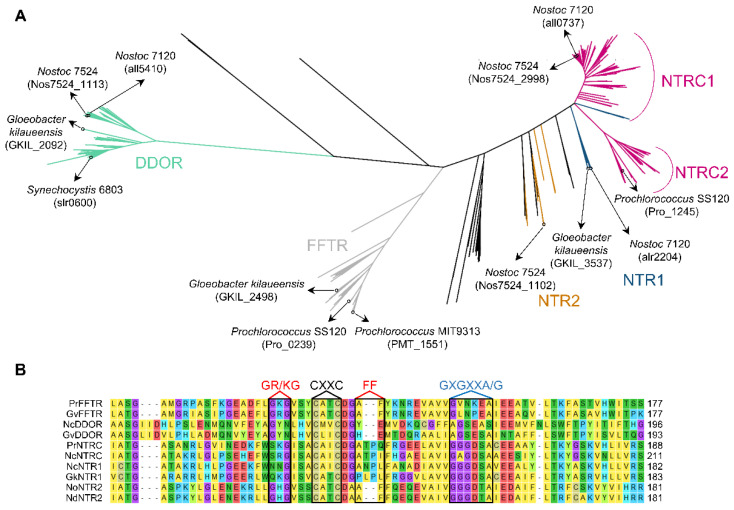
Functional features and evolutionary relationships of TR-related enzymes in cyanobacteria. (**A**) Unrooted tree of amino acid sequences of TR-related enzymes identified in the IMG and NCBI databases. Branches are colored to represent proteins found in the different subclades: DDOR (light blue), FFTR (grey), NTRC (purple), NTR1 (dark blue), and NTR2 (brown). Branches related to *E. coli*, Melainabacteria, Sericytochromatia, and Margulisbacteria are colored black. (**B**) Multiple protein sequence alignment of several TR-related enzymes. Sequences are indicated with the following abbreviations for organism names: *Prochlorococcus marinus* SS120 FFTR (PrFFTR), *Gloeobacter violaceus* FFTR (GvFFTR), *Nostoc* sp. PCC 7120 DDOR (NcDDOR), *Gloeobacter violaceus* DDOR (GvDOR), *Prochlorococcus marinus* SS120 NTRC (PrNTRC), *Nostoc* sp. PCC 7120 NTRC (NcNTRC), *Nostoc* sp. PCC 7120 NTR1 (NcNTR1), Gloeobacter kilaueensis NTR1 (GkNTR1), *Nostoc* sp. PCC 7524 NTR2 (NoNTR2), and *Nodularia spumigena* CCY9414 (NdNTR2). The motifs for Trx-binding (GR/KG and FF) and NAD(P)H-binding (GXGXXA/G) are shown with boxes in red and blue letters, respectively. Black letters indicate the redox-active Cys (CxxC). The full sequences of each TR-related enzyme are available in Appendix A.

**Figure 4 antioxidants-11-00654-f004:**
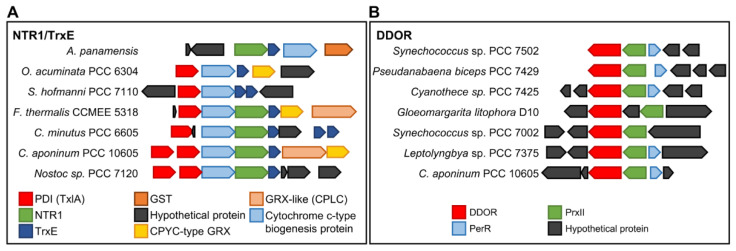
Gene neighborhood conservation of NTR/TrxE (**A**) and DDOR (**B**) from different cyanobacterial genomes. Note that proteins encoded by each gene are indicated.

**Figure 5 antioxidants-11-00654-f005:**
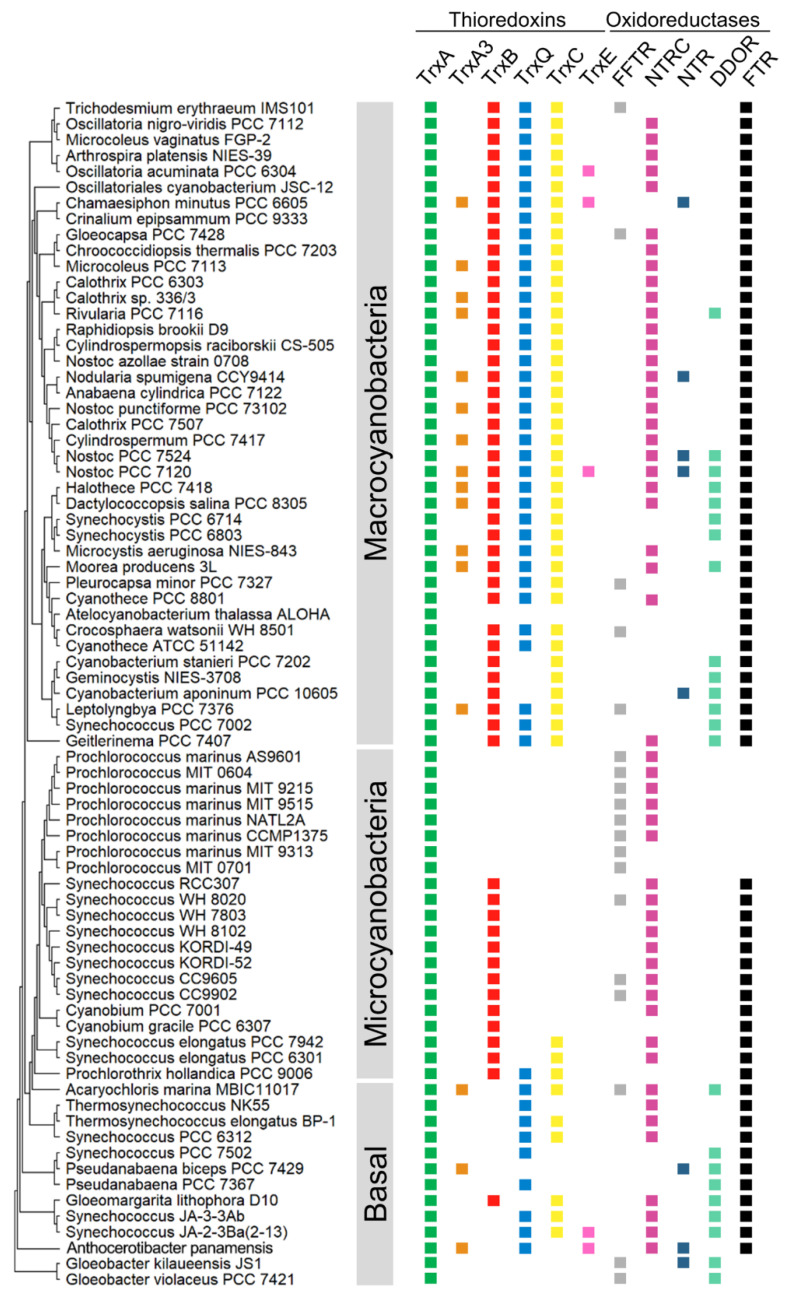
Distribution of Trx systems and related enzymes across the phylum Cyanobacteria. The phylogenetic tree was estimated from 16S rRNA using the maximum likelihood method implemented in MEGA 11.

**Figure 6 antioxidants-11-00654-f006:**
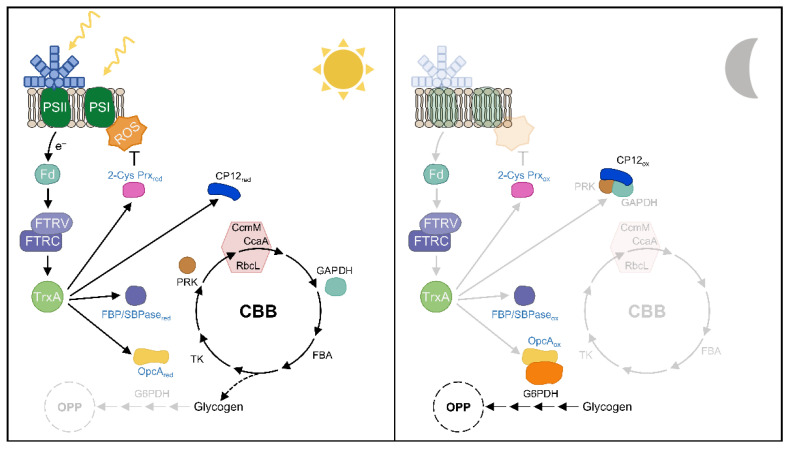
Snapshot of the role of TrxA on target proteins in day–night cycle. Reactions of the CBB cycle, redox homeostasis, and glycogen metabolism are represented. Enzyme names in blue are targets of Trx targets described in vivo. Enzyme names in black are targets of TrxA described in vitro. Abbreviations: PSI and PSII, photosystem I and II; ROS, reactive oxygen species; Fd, ferredoxin; FTRV and FTRC, Ferredoxin thioredoxin reductase variable and catalytic subunit; TrxA, thioredoxin A; G6PDH, glucose-6-phosphate dehydrogenase; OpcA, allosteric effector of G6PDH, FBP/SBPase, fructose-1,6/sedoheptulose-1,7-bisphosphatase; 2-Cys-Prx, 2-Cys Peroxiredoxin; PRK, phosphoribulokinase; CcmM, carboxysome assembly protein M; CcaA, carboxysomal carbonic anhydrase; RbcL, ribulose bisphosphate carboxylase large chain; GAPDH, glyceraldehyde 3-phosphate dehydrogenase; FBA, fructose-bisphosphate aldolase; TK, transketolase; Red, reduced; Ox, oxidized.

**Table 2 antioxidants-11-00654-t002:** Trx target proteins described in vivo. Proteins are indicated with abbreviations for the name of the organism: *Synechocystis* TrxA (SynTrxA), *Anabaena* TrxA1 (AnTrxA1).

Cellular Processes	Target Protein	Regulation In Vivo by	References
CBB cycle	FBP/SBPase	SynTrxA	[15]
OPP pathway	OpcA	AnTrxA1	[29]
Antioxidant defense	2-Cys Prx	SynTrxA	[15]
Transcriptional regulation	PedR	SynFTRV	[84]
Protein synthesis	EF-Tu	SynFTRV	[70]

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
