# Peer review of "Exploring the Diversity of the Thioredoxin Systems in Cyanobacteria"

_antioxidants, 2022, doi:10.3390/antiox11040654_

Round 1

Reviewer 1 Report

Thioredoxins (Trxs) are thiol-disulfide oxidoreductases that regulate the activity of target enzymes involved in many processes, such as photosynthetic CO2 fixation and stress responses. In this manuscript, the authors focus on the evolutionary diversity of Trx systems in cyanobacteria and discuss their phylogenetic relationships. The manuscript is well organized and the language is excellent, and given the present satisfactory presentation of the manuscript, the reviewer considers it ready for publication.

Author Response

Reviewer 1:

Thioredoxins (Trxs) are thiol-disulfide oxidoreductases that regulate the activity of target enzymes involved in many processes, such as photosynthetic CO2 fixation and stress responses. In this manuscript, the authors focus on the evolutionary diversity of Trx systems in cyanobacteria and discuss their phylogenetic relationships. The manuscript is well organized and the language is excellent, and given the present satisfactory presentation of the manuscript, the reviewer considers it ready for publication.

The authors appreciate the interest of the reviewer in the paper and his comments on it.

Reviewer 2 Report

This is a nice review with very interesting updated knowledge and reflexion for future research perspectives in the field of cyanobacteria thioredoxin systems and functions. It is clearly written and illustrated.

I only have some modifications to suggest for improvement...

In the abstract, it should be indicated that Trxs can serve as reducing substrates. This is an important function extensively described in this manuscript.

Along the text:

Line 30-32: ... activated by reduction of one or several disulphide bridge(s) formed between cysteines. In most cases, this reduction is carried out by thioredoxins (Trxs)...

Line 38: ... high diversity...

Line 51: We also included...

Line 109: ... all residues conserved...

Line 116: fulfilled instead of satisfied?

Line 121: FTR should be developed (appearing for the first time in the main text...)

Line 124: since FTRV is by definition variable in size, a range should be reported for the protein size.

Line 159: I don’t understand “NTRC... providing a separate FTR/Trx system”. Do the authors mean a potential alternative Trx system?

Line 165: diverse habitats or diversified habitat.

Line 166-… : Trx reducing systems instead of NTR-related enzymes ? in Fig.3 title, as well... DDOR and FFTR don’t use NADPH as reductant...

Line 201: ...Trx gene downstream NTR1 (ital for genes)...

Line 206: ... allows observing...

Line 326: Subsequently, another Trx...

Line 330: Trxs activate...

Line 332: Furthermore, it was shown...

Line 388: ..., although showing no growth phenotype.

Line 476: ... in the light...

About Figures and tables:

Table1 should not be disrupted in the final version of the ms...

Fig. 1: I found it difficult to distinguish TrxA1 and A2 subclades on the phylogenetic tree (maybe could be indicated on the figure). I wonder why the green branches within black branches in the clade of TrxsA from non-photosynthetic relatives do not form and additional subclade...

Fig.2: ... cyanobacteria analysed. I am not sure to understand the sentence “Cartoon representation indicates position of the Trx models.” Since position of active site Cys residues is shown (should be indicated in the figure caption), the orientation of the models can be clearly visualized. I could not guess what do smaller-sized models correspond to exactly... Are thes cartoons really necessary since active site Cys are indicated?

Fig. 4: ... or DDOR. Genes in ital...

Fig.6: Enzymes names in blue... in black...

Fig. S2: Cite papers sources about amino acids essential for Trx-FTR complex formation...

Fig. S4: ... Indicated... Figure 3B

Author Response

The responses are in the word file
